# Integrating Habitat Suitability and the Near-Nature Restoration Priorities into Revegetation Plans Based on Potential Vegetation Distribution

**Cheng Zheng** [1], **Zhongming Wen** [1,2,*], **Yangyang Liu** [1], **Qian Guo** [1], **Yanmin Jiang** [2], **Hanyu Ren** [1], **Yongming Fan** [1] and **Yuting Yang** [1]

1   College of Grassland Agriculture, Northwest A&F University, Yangling 712100, China; zhengcheng@nwafu.edu.cn (C.Z.); hnlylcbtks@163.com (Y.L.); Guoqian18@nwafu.edu.cn (Q.G.); 1805223632@nwafu.edu.cn (H.R.); fanyongming@nwafu.edu.cn (Y.F.); yutingyang0509@163.com (Y.Y.)
2   Institute of Soil and Water Conservation, Chinese Academy of Sciences and Ministry of Water Resources, Yangling 712100, China; jiangyanmin17@mails.ucas.ac.cn
*   Correspondence: zmwen@ms.iswc.ac.cn

**Abstract:** Selecting optimal revegetation patterns and filtering priority areas can improve the effectiveness and efficiency of revegetation planning, particularly in areas with severe vegetation damage. However, few people include optimal revegetation patterns and priority restoration areas into revegetation plans. The Near-Nature restoration pays attention to "based on nature" ideas, guiding the degraded ecosystems to reorganize and achieving sustainable restoration through self-regulation. In this study, we conducted a field survey of the native vegetation communities in the Yanhe River catchment, and the data obtained were used to construct the potential distribution suitability of the habitat and screen the priority areas through the combination of MaxEnt and prioritizr models. We drew a heat map of species richness by simulating the potential distribution of 60 native species. The results showed that the potentially suitable habitats for forest cover were distributed in the southern part of the Yanhe River catchment; the potentially suitable habitats for herbaceous plant species were located in the center and the northwest parts of the study area; the potentially suitable habitats for shrub plant species in this area were larger than that of the forest, and herbaceous plants species were distributed in many zones of the study area. This study demonstrated that shrubs and herbaceous plant species in parts of the Loess Plateau should be considered as the pioneer plants of revegetation in future revegetation plans. Moreover, we also mapped the priority area of the Near-Nature restoration based on the richness of the potential native species. The procedure followed in this study could provide guidance for revegetation planning and manual management in the regions where vegetation damage occurs.

**Keywords:** MaxEnt; revegetation; habitat suitability; the Near-Nature restoration; prioritizr

## 1. Introduction

Loess Plateau in China (LPC) covers an area of 640,000 km$^2$ and has the deepest and largest loess deposits [1]. For a long time, unreasonable use of the land by local residents, excessive depletion of water resources [2], and climate change have led to serious degradation of the ecosystem and increased soil erosion, which has seriously affected the socio–economic development of the region [3]. Revegetation is one of the effective measures to control soil erosion and improve environmental conditions [4]. In order to combat land degradation and soil erosion in the LPC, the Chinese government has launched several projects at various scales, such as the Shelter-Forest-System Project, Natural-Forest-Protection Program, the Grain-for-Green Program, and the Desertification-Control Program [5–7]. Since the project of the Grain-for-Green Program, aiming to control soil erosion and improve environmental conditions, was initiated, the revegetation cover

has substantially increased [8] and this has effectively controlled soil erosion in this region [9,10]. However, the large-scale conversion of cropland to forest resulted in a decrease in runoff [11] and soil moisture [12,13], as well as an increase in evapotranspiration [1]. There were deviations between species and the environment of the concerned site in the implementation of the policy in the loess hilly region, and the failure to abide by the principle of "restoration according to the demand of water by different species" of revegetation [14], which resulted in excessive water consumption and poses a serious threat to these ecosystems and biodiversity [15,16].

In recent years, the Near-Nature restoration mode has been used in revegetation planning all over the world [17]. It is an effective way that established the model based on Near-Nature to solve the structural instability and the decline of land degradation caused by non-native planting. Near-Nature recovery refers to planting native species in accordance with the vegetation in their natural state through artificial planting [18]. Nevertheless, human activities in many areas of the LPC have caused serious damage to natural vegetation, such as in the Yanhe River catchment (Figure 1) [19]. It is not easy to select natural vegetation for the Near-Nature restoration plans. In the past few years, many researchers have simulated the potential vegetation distribution in the area through a variety of models [20–22]. Wen et al. [21] used a generalized additive model to simulate the distribution of 37 plant communities in the study area and the potential vegetation distribution in the catchment. Peng et al. [22] established the LPJ-GUESS model to construct potential vegetation distribution based on a physically-based ecosystem. Nonetheless, these models mainly require large samples of the data and perform poorly in explaining the habitat suitability of species, as well as the habitat suitability and restoration potential of the native species being rarely taken into consideration in vegetation restoration plans in the previous studies. Therefore, very little information is available about which species are used for vegetation restoration and which areas should be prioritized for the Near-Nature restoration based on the restoration potential of native species. Geographic Information Systems (GIS) and machine learning have been widely used in the field of agricultural meteorology. The MaxEnt model has become an important tool for suitability zoning due to its objective, quantitative characteristics and good performance. This model has been successfully used in the habitat simulation of a variety of cash crops, medicinal materials, and invasive alien species [23–25]. Priority restoration is an effective way to improve the efficiency of vegetation restoration. Several prioritization methods have been developed around the world [26–29]. The prioritizr R package uses integer linear programming (ILP) techniques to provide a flexible interface for building and solving planning problems [30]. This package has the functionality to read input data formatted for the Marxan conservation planning program and can find much cheaper solutions in much shorter time than Marxan. Such a tool could also help facilitate a revegetation decision-making framework, thus enhancing prioritization of revegetation efforts in multiple landscapes.

The prioritizr R package, a tool of systematic planning and decision, was applied to identify the priority areas for revegetation based upon MaxEnt models. It focuses on the planning of the revegetation system in the Yanhe River catchment, where the focal area of revegetation in LPC is located. The model took multiple native species as the benchmark to divide the habitat suitability level based on climate and topography as a unit (500 m*500 m) for planning restoration priorities. The goal of the paper is to (a) examine the geographical distribution of native species; (b) discuss the habitat suitability of different vegetation types; (c) draw a heat map of 60 native species according to native species potential distribution by MaxEnt; (d) recommend prioritization of the Near-Nature recovery areas based on the natural species for future effective revegetation. This model will be an important supplement to the Grain-for-Green program. The methods and results of this study may provide a basis for planning in the areas undergoing similar revegetation and help develop sustainable vegetation management plans.

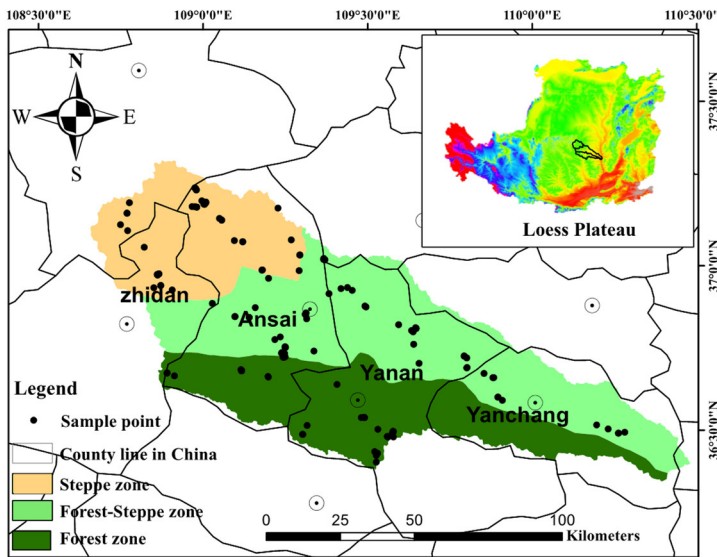

**Figure 1.** The locations in Yanhe River catchment in the Loess Plateau in China (CLP). Each dot is a sample plot, and a total of 145 observation plots were used to establish distribution models.

## 2. Materials and Methods

### 2.1. Study Site

The Yanhe River catchment covers an area of 7687 km$^2$ and is located in the middle of the CLP (36°23′ N–37°17′ N, 108°45′ E–110°28′ E), including Zhidan, Ansai, Yan'an, and Yanchang from the northwest to southeast (Figure 1) [22]. The catchment has a continental climate, with an annual rainfall of 420~540 mm and an average annual temperature of 5–13 °C. The area of loess hills and gully in the basin accounts for 90% of the entire basin, and the average slope is 4.5%. Meanwhile, the terrain in this area is highly fragmented, and the terrain has large undulations [31]. The terrain has a strong redistributing effect on climatic factors such as rainfall and temperature, which determines the complexity and diversity of the ecological habitats in the studied area. The dominated vegetation covers are the grasslands in the northwest; the forest-cover increases gradually from the northwest to southeast, which is mostly located in the Yanchuang area and the south part of the Yan'an area. This catchment is the increasing focal area of the vegetation restoration and reconstruction in the LPC due to the serious vegetation destruction and the fragile ecological environment [32].

### 2.2. Occurrence Database and Environmental Variables

A total of 60 native vascular plant species in the region were identified, and the coordinate information of these species' occurrence was collected using Global Positioning System (GPS). The details are provided in the Supplementary Materials (S1). We obtained 145 occurrence plots from the northwest to southeast of the Yanhe River catchment based on the environmental stratified sampling method [21]. We processed the data collected in the field, and kept the samples of natural vegetation distribution. In order to reduce the sampling deviation and eliminate the spatial autocorrelation, we also used the spThin package from R to analyze the data and remove the duplicated data of the sample square in the grid with a unit of 25 m.

According to existing studies [21,33], we first selected nine climatic factors and four topographic factors related to plant growth, including temperature, precipitation, and topography, as environmental data to simulate the distribution of species in the Yanhe River catchment, which are biologically significant to define physiological and ecology limits of a plant species [30,33,34]. The meteorology data were obtained through spatial interpolation of data from 57 weather stations by ANUSPLN [35]. A Digital Elevation Model (DEM) was generated from the 1:10,000 topographic map of the Yanhe River catchment using

ARC/INFO from ArCGIS. Among the terrain factors, slope, aspect ratio, and elevation were calculated [36]. This study adopted the residual elevation analysis and combines the slope to divide the terrain parts of the Yanhe River catchment into seven categories, namely, flat land between river channels and ditch, downslope, mid-slope, upslope, ridge top, and high flat land [21]. The description of environmental factors is shown in Table 1. We executed some checks prior to the formal analysis and eliminated climate factors with low contribution rates which were less than 5% and similar properties to avoid overfitting of environmental factors. Finally, we obtained environmental factors to simulate the distribution of native species in the Yanhe River catchment.

**Table 1.** Environmental variables that were used to build the native species distribution models using MaxEnt and description of these variables.

| Abbreviation | Environment Variables | Range | Unit | Abbreviation | Environment Variables | Range | Unit |
|---|---|---|---|---|---|---|---|
| TemCM | average monthly temperature of the coldest month | 18.99–26.55 | °C | RainA | average annual rainfall | 420.72–439.41 | mm |
| TemHM | average monthly temperature of the hottest month | −5.26 | °C | RainRA | average annual rainfall in rain season | 243.73–314.42 | mm |
| Tem | average annual temperature | 5.75–12.60 | °C | RainSEA | average annual rainfall seasonality | 0.51–0.59 | mm |
| Temsea | seasonal temperature change ratio | 1.64–1.78 | °C | Eva | Annual average evaporation | 809.73–1032.18 | mm |
| TemAO | average monthly temperature (April to October) | 13.21–20.56 | °C | Elev | elevation | 495–1795 | m |
| SloD | slope degree | 0–66.14 | °C | SloP | slope position | 1–7 | - |
| SloA | slope aspect | 1–5 | °C | - | - | - | - |

### 2.3. Setting of MaxEnt Software Parameters

The calculation of plant distribution in the Yanhe River catchment by the MaxEnt model 3.4.1 Version, downloaded from https://biodiversityinformatics.amnh.org/open_source/maxent/ [37]. MaxEnt model relies on the geographical data collected in the field to make a comparison of the sampled environment variables of the reference set of grid cells with those representing the observed presence of the species [38]. The MaxEnt model required at least five different occurrence data for each species to obtain more accurate results [37,39]. As a consequence, we also used it as the minimum standard for calculating species distribution. The "random test percentage" was 25 (representing a random selection of 75% of the distribution points as the training dataset to build the model, leaving 25% of the distribution points as the test dataset); the model repeatedly runs ten times under the same setting and each run is considered as a single replication; "subsample" was selected as the "replicated run type", and the final result took ten repetitions (on an average) [40]. The output format is set as "Logistic" probability distribution, the threshold segmentation is determined by "the lowest presence threshold (LPT)" method.

The accuracy of the algorithms to make correct predictions was assessed through two parameters, i.e., the area under Receiver Operating Characteristic (ROC) curve (AUC) [37] and true skill statistic (TSS) [41]. The value of AUC varied between 0 and 1, among which, the standard is 0.9 < AUC < 1.0, highly accurate; 0.8 < AUC < 0.9, good; 0.7 < AUC < 0.8, useful [42]. TSS varies from −1 to +1, where +1 stands for excellent agreement, and a value of ≤ 0 indicates that the performance is not superior to random predictions.

### 2.4. GIS and Priorities Analysis

We made the visible map of species abundance using ArcGIS 9.3. The range of possibilities for species was between 0and 1; the lowest presence threshold (LPT) was

used to define the suitable and unsuitable habitats of single species [39]. We obtained the number of species in each grid to express species richness by superimposing all the species distribution layers from MaxEnt output using the grid calculator function of ArcGIS 9.3 [43].

The higher the richness of potentially distributed species, the higher the restoration potential of the area. In order to facilitate the restoration of the study area and refer to the actual topography of the study area, we set up a 500 × 500 m square fishnet for the study area for restoration planning and a total of 54,415 planning units to formulate a restoration plan for the study area. We used the min_objective and max_features of the prioritizr R package to describe the Near-Natural restoration priority of the Yanhe River catchment. The prioritizr R package is similar to the Marxan decision support tool in that we can generate solution combinations and calculate the frequency of planning unit selection to understand their relative importance. In addition, we created a portfolio containing 1000 solutions within 20% of optimality and calculated the number of times that each planning unit is selected.

## 3. Results

### 3.1. Model Assessment

As shown in Table 2, the simulation of the distribution of 60 native species in the Yanhe River catchment was acceptable considering all accuracy measures considered (AUC > 0.7 and TSS > 0.1; Table 2); thus, the resulting potential distributions of the 60 native species in the Yanhe River catchment were considered to provide a reliable estimate of the impact of suitability prediction.

**Table 2.** Comparison of area under the Receiver Operating Characteristic (ROC) curve (AUC) and true skill statistic (TSS) of the MaxEnt model.

| AUC/TSS | 0.7–0.8/0.1–0.2 | 0.8–0.85/0.2–0.4 | 0.85–0.9/0.4–0.6 | 0.9–0.95/0.6–0.8 | >0.95/0.8 |
|---|---|---|---|---|---|
| Area under the Curve (AUC) | 1 | 23 | 16 | 16 | 4 |
| True Skill Statistic (TSS) | 2 | 13 | 27 | 15 | 3 |

The previous studies on model comparison have shown that the prediction accuracy of MaxEnt was better than that of other models, especially in the case of unknown species distribution [37,44]. The model was repeated ten times to obtain the mean value and output results, which can reduce the uncertainty of the results. Our MaxEnt model seems to predict the distribution of species on the Yanhe river catchment very well, as 33% of the species performance was good (AUC > 0.90), and only 3% of them were useful (AUC < 0.80) [6,42,45]. However, our study results have 15 species with 0.1 < TSS < 0.4, indicating that the distribution of these 15 species by MaxEnt is not completely reliable. We observed a significant negative relationship between the number of geographical distribution points and mean AUC for the native species ($n = 60$, $r = -0.50$, $p < 0.01$), which was similar to the observations made by Alatawi et al. [46]. Although AUC does not seem to be significantly correlated with different vegetation types, most of the species with AUC > 0.9 are tall semi shrubs. This observation also reflects the importance of semi shrubs in the study area.

### 3.2. The Potential Distribution of Species and Influencing Factors

We simulated the potential distribution of 60 species of the Yanhe River catchment. Here, we only presented four common species that were used for revegetation, including *Thymus mongolicus* Ronn., *Artemisia sacrorum* Ledeb., *Ostryopsis davidiana* Decaisne., and *Quercus wutaishansea* Blume., which play an important role in revegetation. The species distribution was indicated in Figure 2. Thymus mongolicus Ronn. was mainly distributed in the northwest of the study area, and its distribution probability increased along the northwest direction. *Thymus mongolicus* Ronn. is more dispersed than the other three species. *Ostryopsis davidiana* Decaisne. was mostly scattered in the south of the study area,

and the distribution probability was the highest in the southern parts of the study area. The model predicted that the climate in the temperate and subtropical regions of Central and Northwest Yanhe River catchment is suitable for the growth of *Artemisia sacrorum* Ledeb. *Quercus wutaishansea* Blume. was largely distributed in the southern parts of the study area, and the probability of distribution increases toward the geographic south direction. Moreover, MaxEnt predicted that the potential geographic distribution with high suitability has the largest area compared to the others.

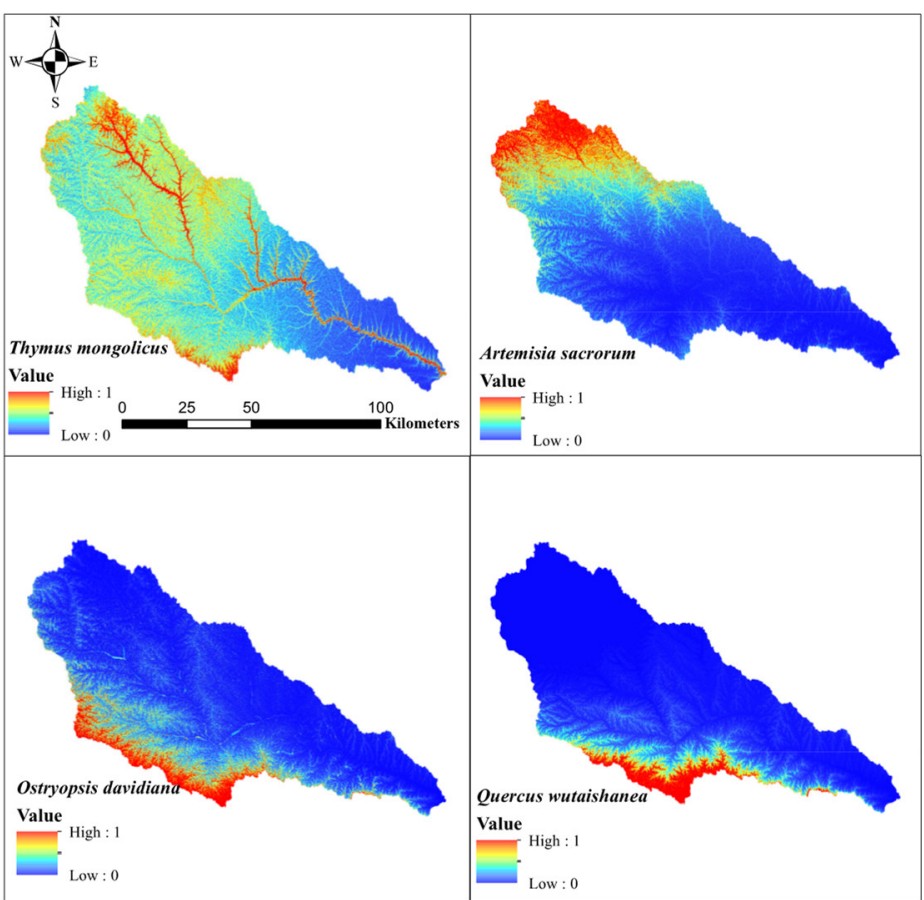

**Figure 2.** Potential geographical distribution of four typical plants on the Yanhe River catchment as predicted by MaxEnt.

Jackknife tests analyzed in MaxEnt on the environmental variables indicated that seasonal temperature change ratio (Temsea, 37%) and slope degree (SloD, 12.3%) were the most important environmental factors affecting the distribution of *Thymus mongolicus* Ronn. and their cumulative contribution rate was as high as 49.3%. Seasonal temperature change ratio (Temsea, 67.5%) was the most important environmental factor in the distribution modeling of *Ostryopsis davidiana* Decaisne. Among the several environmental variables screened for *Artemisia sacrorum* Ledeb., the contributions of annual average evaporation (Eva, 54.3%) and average annual rainfall (RainA, 22.3%) were much larger than those of other environmental factors. The most important variable for *Quercus wutaishansea* Blume distribution was the average annual rainfall (RainA, 62.2%) (Table 3).

**Table 3.** Environmental variables and their contributions to the distribution of four native species (the environmental variables that contribute less than 5% to the distribution of the four local species are not shown here).

| Native Species | Percent Contribution | | | | | | | | |
|---|---|---|---|---|---|---|---|---|---|
| | Temsea | RainA | TemCM | RainSEA | RainRA | SloP | Eva | SloA | SloD |
| *Thymus mongolicus* | 37 | 7.1 | 11 | 7.8 | <5 | <5 | <5 | <5 | 12.3 |
| *Artemisia sacrorum* | <5 | 22.3 | <5 | <5 | <5 | <5 | 54.3 | <5 | 5.1 |
| *Ostryopsis davidia* | 67.5 | <5 | <5 | <5 | <5 | 5.6 | <5 | 9.6 | <5 |
| *Quercus wutaishansea* | 9.9 | 62.2 | <5 | <5 | 12.1 | 6.9 | <5 | <5 | <5 |

During the analysis of the geographical distribution of native species and environmental variables, we screened those with the environmental variable contribution rate greater than 5% for the analysis. We also divided the environmental variable contribution rate into four gradients (5–15%; 15–30%; 30–45% and >45%). As shown in Figure 3, Temsea is the highest (48 species), and the contribution value of the geographical distribution of 36 native species is more than 15%, indicating that Temsea plays an important role in the geographical distribution of the native species. SloD is the second most important factor affecting the geographical distribution, and it contributes to more than 5% of the geographical distribution of 43 native species and occupies an important position in the distribution of 11 species (>15%). Average annual rainfall seasonality (RainSEA) and rainfall in rain season (RainRA) contribute less to the model, while the contribution rate of environmental factors other than the icon display is less than 5%. In general, the Temsea and SloD have greatly contributed to the occurrence of native species in the Yanhe River catchment.

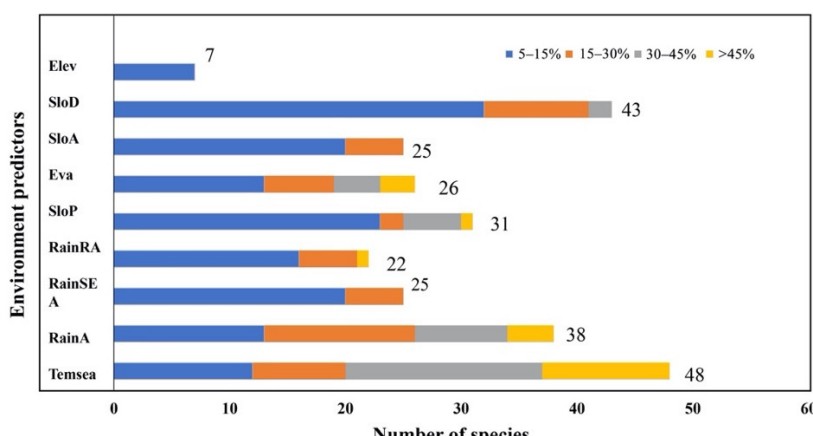

**Figure 3.** The frequencies of each environmental predictor and their contribution to the models of 60 main species in the Yanhe River catchment. The variable contributions were assessed using permutation importance in the MaxEnt output.

### 3.3. Native Species Richness and Different Vegetation Types in the Yanhe River Catchment-Based on Potential Vegetation Distribution

It was observed that the hot spots in the Yanhe River catchment were mainly distributed in the south of the study area, most of the river valleys in the Ansai area, and part of the Yan'an area. It occupied about 1529 km$^2$ and accounted for 20% of the study area. The species richness of 60 common plant species in the Yanhe River catchment was shown in Figure 4 and Table 4. The species richness is the lowest in the eastern part of the study area, and the highest species richness distributes in the valley, covering an area of 715 km$^2$.

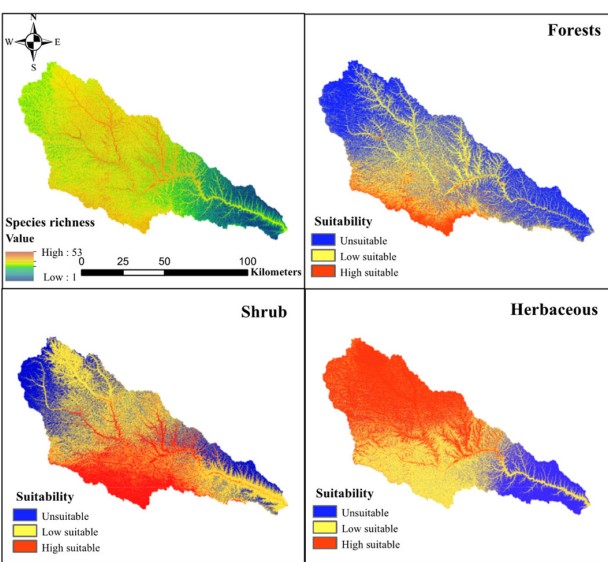

**Figure 4.** Vegetation distribution hotspots in the Yanhe River catchment.

**Table 4.** The distribution area of species richness hot spots in the Yanhe River catchment (the first column represents various levels of species richness, the second column shows the corresponding area occupied, and the third column is the percentage of the distribution area in terms of the study area).

| Species Richness | Area (km$^2$) | Number (%) |
|:---:|:---:|:---:|
| 0–14 | 715.61 | 9.33 |
| 14–25 | 920.76 | 12.1 |
| 25–33 | 1768.4 | 23.1 |
| 33–39 | 2723.45 | 35.56 |
| 39–53 | 1529.9 | 20.01 |

As indicated in Figure 4, we divided the investigated plants into three vegetation types: herbaceous, shrub, and forests. We also classified the research area into highly suitable areas (>50%), low suitable areas (>20%), and unsuitable areas (<20%) according to species richness to explore the suitability response of different vegetation types to the habitat in the study area. The highly suitable area for forests is mainly scattered in the southern part of the study area, and the low suitable area is located in the Yan'an area. The suitable distribution area of forest is the lowest among the three vegetation types. The highly suitable area for shrubs is mainly located in the middle and southern part of the study area. Shrubs have the widest suitable area when compared to other vegetation types in the study area. Herbaceous plants have the largest distribution of highly suitable areas distributed in the middle and northern part of the study area. However, the overall distribution area of suitable areas is smaller than that of shrubs.

*3.4. Priority Near-Natural Restoration Area Identification Based on Prioritizr*

The prioritizr R package was set as the restoration priority zone was based on the potential species richness of native species along the Yanhe River catchment. According to the potential species richness of native species along the Yanhe River catchment, the priority of restoration was determined. We made a portfolio of 1000 solutions with optimality within 20%, and assured that each planning unit was selected. The priority zone recognition based on the description of the min-objective solution is shown in Figure 5a,c and the priority zone recognition based on the description of the max_features solution is shown in Figure 5b,d, whose solutions for the two problems had high similarity. In addition, we also identified the priority area based on removing the species with TSS < 0.4 (Figure 5).

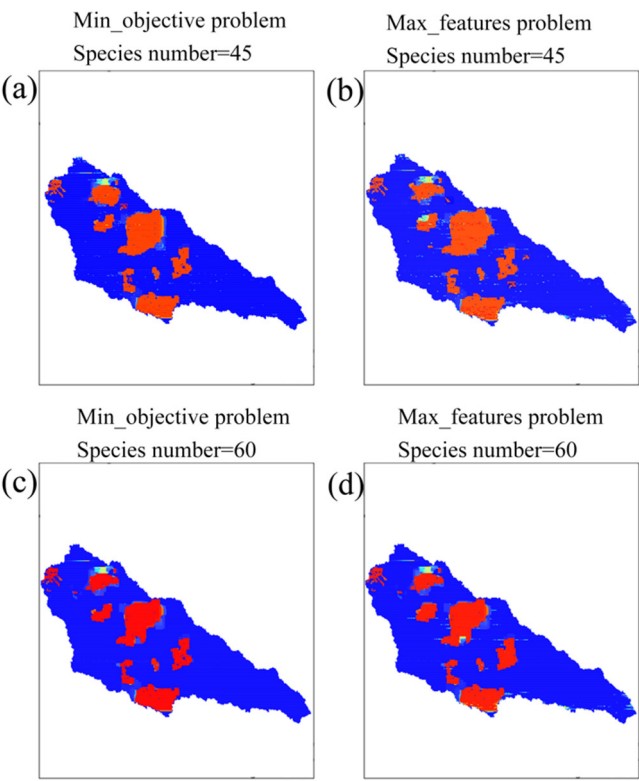

**Figure 5.** Vegetation restoration priority area by prioritizr for different, species number and problem solutions. Across all panels, filled in red grid cells represent selected planning units for all the time period (1000 times), filled blue in grid cells represent not selected planning units (0 times). (**a**) Min_objective problem, species number = 45; (**b**) Max_features problem, species number = 45; (**c**) Min_objective problem, species number = 60; (**d**) Max_features problem, species number = 45.

## 4. Discussion

### 4.1. The Potential Distribution of Species and Influencing Factors

Predicting the suitable habitats for the native species in the Yanhe River catchment is critical for developing a revegetation plan [47]. Although there were many models to simulate the potential distribution of species [21], this study can provide a high-precision species model of a large number of species by exploring the relationship between species and environmental heterogeneity and provide a method for comprehensively understanding the vegetation status of the area. In the present study, species occurrence data were obtained from the field investigation and had extremely high accuracy. In addition to considering the impact of meteorology data on species distribution, this paper also combined with the actual complex topographic factors in the LPC, which exert an indirect effect on species distribution. According to Qin and Liu, the redistribution of resources by topographic factors in the LPC affected species distribution, and our results were also consistent with their observations [48,49]. Our results have also shown that slope aspect (SloA) and SloD significantly affected more than 50% of the 60 species studied (Figure 3). Different topography results in differences in the nutrients content and moisture in the soil, which may be one of the reasons for the difference in species distribution due to diverse topography [50].

Determining the environmental variable that is shaping and maintaining a species' geographical occurrence is a key issue in selecting the species for revegetation. We discussed the contribution value of each environmental factor to the potential distribution of individual species and attempted to explain the relationship between species distribution and the environment through niche theory. For example, the average annual temperature showed a negative nonlinear response, but a positively nonlinear response was observed

for annual average evaporation in Artemisia sacrorum (Figure 6). The main reason why the researchers chose meteorological and terrain factors was that temperature, light, and moisture are the most decisive factors affecting the distribution of plant species. Secondly, temperature and moisture have large-scale, long-term, and stable observational data and, therefore, this method is adopted by most scholars. Thirdly, the topography of the Loess Plateau is complex and changeable, and it has the ability to redistribute temperature, light, and water [51,52]. Nevertheless, other factors also affect the distribution of the native species. For example, soil factors play an important role in the regional distribution of native species [53]. In studies with a small scale area, canopy opening, and corridor landscape such as a river also influence the native species distribution [54,55]. Full consideration of multiple indirect factors and in-depth disclosure of key factors affecting the distribution of different species in the future will help improve the accuracy of simulation results.

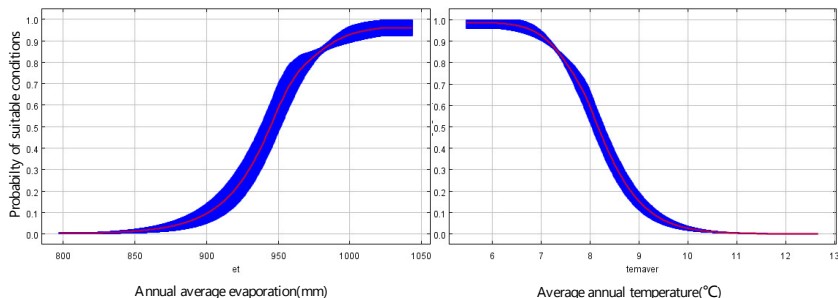

**Figure 6.** Average response curves of the main predictor variables of the modeled distribution of *Artemisia sacrorum* on the MaxEnt algorithm.

*4.2. Habitat Suitability of Native Species Based on Potential Vegetation*

We investigated for the existing natural vegetation in the study area, investigated the species composition in the natural vegetation, and constructed the distribution pattern of the potential vegetation in this study. The research results indicated that the herbaceous species were distributed in the central and northern parts of the study area. The potential distribution area of shrubs can almost cover all the study areas, and the potential distribution area of forests is located in the southern part of the study area. The results of the study were similar to those of previous works [56]. Since 1999, large areas of trees have been planted in this area, which is contrary to our proposed regional vegetation restoration plan [11]. We hold the view that large numbers of *Robinia pseudoacacia* are planted in some unsuitable areas, and that may cause the soil water level to drop, and the sustainability of the vegetation restoration may deteriorate [19,22,57]. *Robinia pseudoacacia* is one of the most widely introduced species in the world. It is also widely planted in the LPC. It is the main tree species for artificial afforestation in the Loess Plateau [19]. *Robinia pseudoacacia* does not match the site environment in the hilly area, which leads to excessive consumption of soil water, adverse effects on water resources and vegetation diversity, and negative phenomena such as biomass overload [2]. Our study shows that the suitable area of herbs and shrubs is greater than that of trees in the Yanhe River catchment. In the future vegetation restoration plan, we could try to use shrubs and herbs for vegetation restoration. This would improve the effectiveness and efficiency of vegetation restoration in this area.

While researching the plant habitat suitability models, most scholars only use climatic factors to establish models [58–60]. This study combines climatic factors and topographic factors for model simulation and found that for small-scale vegetation habitat suitability studies, the slope degree may also be an important factor that needs to be adopted, and this factor has been seldom considered in the previous studies. The MaxEnt model's output shows that RianA, TemSEA, and SloD are the key factors restricting the distribution of native species in the Yanhe River catchment. The degree of influence of the SloD may be greater than that of RianA; the species richness simulated by the MaxEnt has shown that the regions with higher richness are distributed in the southern and central valley

regions, which are suitable habitat areas for the native species [61]. The species richness was low in the southeastern part of the study area, but the population density in this region is relatively high. Human activities have affected the natural vegetation resulting in fewer sampling points. Moreover, the southeastern part of the study area is distributed with forests and has low herbaceous species diversity.

The habitat suitability of the 60 native species in the Yanhe River catchment was added together to obtain the distribution pattern of habitat suitability in the study area. Compared with the previous single species to construct habitat suitability, the deviation of potential habitat suitability evaluation is improved. This study quantified the habitat suitability of the study area. It provides useful information for the protection of biodiversity and vegetation restoration in the study area.

### 4.3. Identification of Priority Areas for the Near-Nature Restoration

Some recent studies have shown that the substitution of native species with alien species may cause loss of diversity among forest species [62]. In recent years, the concept of "Near-Natural restoration" has received widespread attention from ecologists all over the world [63]. Near-natural restoration relies on natural ecological processes to restore the degraded ecosystem to an ecosystem with species composition, diversity, and community structure close to zonal communities [64]. This process also aims to achieve the diversity, stability, and sustainability of the ecosystem structure and function after restoration. Therefore, the restored ecosystem has higher biodiversity, can provide more ecosystem functions and services, and increased resilience to disaster risks. Zhang et al. [65] planned the priority conservation areas for the plant species of Yunnan, China, using both MaxEnt and Marxan, and provided an important reference for our work. We described potential changes in the geographic distribution and occurrence probability of the native plant species, and this information was used to identify priority areas for revegetation and management recommendations within the study area. The higher the abundance of potential suitable native species, the higher the suitability of native species in the area, and the stronger the natural restoration capacity of vegetation restoration works.

Nevertheless, this study still has certain limitations in the identification of Near-Natural priority areas. This investigation employed the area of the planned unit of the study area as the cost data. However, the current restoration cost obtained only by the size of the area makes the result uncertain resulted from the complexity of the regional environment and the limitations of the survey. Therefore, improving regional environmental data, land use type cost calculation and population density survey will be of great significance to regional vegetation restoration planning. Secondly, the area of the planning unit will affect the planning results. This study area is divided according to the plant environment hierarchical sampling unit, considering the differences in the meteorological environment of different units and affecting the distribution of different species. Researchers who will apply this model in the future need to use it carefully and set it according to different recovery requirements.

### 5. Conclusions

In this study, the priority area for the Near-Nature restoration of the LPC was determined by evaluating the species richness of potential native vegetation, which provided a reference for the effectiveness of the revegetation plan. The area of the priority zone is mapped for reference This study also proved that the habitat suitability of the native species in the Yanhe River catchment was mostly affected by seasonal temperature change and slope degree. The seasonal temperature change rate contributes more than 5% to the distribution of 48 native species. The contribution rate of distribution exceeds 15%, and it occupies a dominant position in determining the distribution of 36 native species. The slope degree contributes more than 5% to the geographical distribution of 43 native species and occupies an important position in the distribution of 11 species (>15%). In potential vegetation simulation and prediction, the slope degree was as important as other climatic

factors. Furthermore, shrubs were most widely distributed in the suitable areas of the study area; herbaceous was most widely distributed in the highly suitable areas of the study area. In some areas of the LPC, shrubs are more suitable for vegetation restoration than forests. This approach of the Near-Nature restoration priority area planning could improve revegetation's effectiveness and mitigate vegetation degradation.

**Supplementary Materials:** The following are available online at https://www.mdpi.com/1999-4907/12/2/218/s1, Text S1: Description of methods for potential vegetation distribution based on MaxEnt model, Table S1: Number of sampling plots and environmental contribution ratios of 60 native species in the Yanhe River catchment.

**Author Contributions:** C.Z., designed the study, conducted the statistical analysis and drafted the manuscript. Z.W., proposed the idea, wrote the protocol and supervised all work. Y.L. and Q.G., edited the manuscript and managed literature searches. Y.J., H.R., Y.F. and Y.Y., carried out data and specimen collection. All authors read and approved the final manuscript.

**Funding:** This research was funded by the National Natural Science Foundation of China (NSFC) (41977077, 41671289).

**Acknowledgments:** This study was supported the National Natural Science Foundation of China (NSFC) (41977077, 41671289). We would like to give thanks to the anonymous reviewers for their meaningful comments on our manuscript.

**Conflicts of Interest:** The authors declare no conflict of interest.

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
