# Peer review of "Integrating Habitat Suitability and the Near-Nature Restoration Priorities into Revegetation Plans Based on Potential Vegetation Distribution"

_forests, doi:10.3390/f12020218_

Round 1

Reviewer 1 Report

Dear authors,

I have provided many comments and suggestions, which I hope that you will find useful, constructive and to the betterment of your manuscript.

Author Response

Dear Reviewers:
Thank you for the reviewers’ comments concerning our revised manuscript entitled “Integrating habitat suitability and the Near-Nature restoration priorities into revegetation plans based on potential vegetation distribution” (ID: forests-1059230). On behalf of my co-authors, we thank you very much for giving us an opportunity to revise our manuscript. We have made significant revisions to our manuscript to address the comments. Those comments are all valuable and very helpful for revising and improving our paper, as well as the important guiding significance to our research. We have studied these comments carefully, and have made corrections which we hope meet with approval. All modifications in the revised manuscript have been highlighted as red font.
Thanks again for the opportunity for allowing us the chance to revise our manuscript. We would also like to thank the reviewers for their time. The main corrections in the paper and the responds to the reviewer’s comments are as following. Please see the attachment.

Reviewer 2 Report

Overall, the manuscript is relevant to restoration practices and could be of high value to the ecological community. Tables and Figures help explain key concepts of the research. There are multiple key ideas and concepts to be clarified, particularly within the methods and rationale for decision making. Some of the decisions, as currently presented, seem arbitrary or without clear scientific reasoning, and those must be justified clearly in the manuscript. Additional data would also be useful (either within the manuscript or supplementary materials).

Line 7: Northwest should be Capitalized

23: "Demonstrated" is probably a better term to use than "proved"

25: If space is available in the abstract, a brief description of Near-Nature would assist readers not familiar with the term.

29: "species richness" should be two terms.  I would also consider "revegetation" as a keyword.

38: "Grain-for-Green Program" could use some more historical detail. Also quotation mark after program is in incorrect location.

39: "invegetation" needs space.

49: "actually" not necessary here

58: Need more information on "habitat suitability and restoration potential of the native species" as to why these models could be improved.

71: Need more detail about why combining MaxEnt and Marxan is the best approach. Why not just use Marxan? No clear evaluation of why the more complicated combination would yield better results.

76: objective C needs clarification. What is meant by "the species richness?"

77: "priorities" or "prioritization" seems a better term

96: 60 native vascular plant species, it seems.

97: How were these 145 points determined? Random? Representative?

99: What is meant by "enclosed environment"? Also "kept" instead of "keep"

102: "sample square in the grid with a unit of 5 m" needs further discussion.  Overall, it is very unclear how species were collected for this study, which makes it hard to evaluate the utility of the data.

106: Supplemental information needed about models evaluating contribution rates of climate factors, such that it is clear why those factors were eliminated.

110: This paragraph is a little difficult to follow. Should mention how data were acquired first and then discuss eliminating those that were low contribution.

116: Figure 1 Legend "Mainly County" seems confusing. Different term would make sense. 118 and 119 seem to provide clarification, but numbers are difficult to read.

121: Table 1 is difficult to interpret. Re-structure for clearer meaning.

127: "and so on" seems out of a place. A bit informal.

137: "sea of native species" seems an odd word choice

151: "ran" instead of run

163 and 172: space needed in scientific name

167: What ties together the species in the different categories of AUC value? Are they more herbaceous, forest, etc.? Particularly those >0.9, as those would make the most sense to prioritize in restoration. Supplemental information includes this info, but I believe it's relevant to the study.

168: Table should read 0.9-0.95 instead of 0.09-0.95

171: Why were those typical species chosen? They weren't the most common or most unique. Also, much of this paragraph seems to be more relevant for the methods.

180: sacrorum should be lowercase.

213: Figure 3 yaxis needs to be clearer.

222-225: terminology should be consistent and not simply re-stating table 3.

270-282 are results, not discussion.

290: meteorological should be lowercase

Figure 6 should be results, not discussion

312: further discussion of historical actions and future ramifications would be helpful here

324: extra space before which

342-348: This should all be in the introduction

355: Need more discussion of impacts of the results, not just restating the methods and results

References: Some formatting issues (Journal Titles Abbreviated vs Full Words; some journal titles capitalized, some volume numbers missing, etc.).

Author Response

Dear Reviewers,

Thank you for the reviewers’ comments concerning our revised manuscript entitled “Integrating habitat suitability and the Near-Nature restoration priorities into revegetation plans based on potential vegetation distribution” (ID: forests-1059230). On behalf of my co-authors, we thank you very much for giving us an opportunity to revise our manuscript. We have made significant revisions to our manuscript to address the comments. Those comments are all valuable and very helpful for revising and improving our paper, as well as the important guiding significance to our research. We have studied these comments carefully, and have made corrections which we hope meet with approval. All modifications in the revised manuscript have been highlighted as red font.
Thanks again for the opportunity for allowing us the chance to revise our manuscript. We would also like to thank the reviewers for their time. The main corrections in the paper and the responds to the reviewer’s comments are as following. Please see the attachment.

Round 2

Reviewer 1 Report

Dear authors,

Thank you for your detailed response to all of my comments. I have made some suggestions in your new manuscript that you need to address. Overall, you greatly improved your text.

Author Response

Dear Reviewers:                                                                                       

Thank you for  for the reviewers’ comments concerning our revised manuscript entitled “Integrating habitat suitability and the Near-Nature restoration priorities into revegetation plans based on potential vegetation distribution” (ID: forests-1059230). On behalf of my co-authors, we thank you very much for giving us an opportunity to revise our manuscript. We have made significant revisions to our manuscript to address the comments of reviewers. Those comments are all valuable and very helpful for revising and improving our paper, as well as the important guiding significance to our research. We have studied these comments carefully, and we have made corrections which we hope meet with approval. All modifications in the revised manuscript have been highlighted as red font.

Thanks again for the opportunity for allowing us the chance to revise our manuscript. We would also like to thank the reviewers for their time. The main corrections in the paper and the responds to the reviewer’s comments are as following.Please see the attachment.

Reviewer 2 Report

Much improved revision with clear addressing of most of the previous comments.

Some key details of methods, which should be described in MS, are instead in the Excel file of the supplementary materials, which makes it more difficult for interpretation. A clear explanation would help.

Table 3 scientific name spacing is still inconsistent

Still quite a few English typos (for example, figure 6 alone has several; "solition" instead of "solution" and "selectction" instead of "selection"); full MS needs further English evaluation

Author Response

Dear  Reviewers:  

We are grateful to you for your effort reviewing our paper and your positive feedback. Here below we address the questions and suggestions raised by the you.

Round 3

Reviewer 1 Report

Dear authors,

I've made some suggestions on your manuscript that you should consider addressing. As stated in previous review rounds, please provide better figures and more descriptive figure legends. Also, as a sensitivity test, please consider running your analyses with and without the 15 species that have very low AUC & TSS scores. Finally, please consider formulating at least one more problem with prioritizr in order to compare your results - which as they stand are similar to Marxan prioritization schemes - with the results obtained e.g., from the max_features function from prioritizr. 

Author Response

Dear  Reviewer:   We are grateful to reviewer #1 for your effort reviewing our paper and your positive feedback. Here below we address the questions and suggestions raised by the reviewer #1. Please see the attachment.  

Round 4

Reviewer 1 Report

Dear authors,

I have no more comments to make.